# Prognostic Value of Chemotherapy Response Score (CRS) Assessed on the Adnexa in Ovarian High-Grade Serous Carcinoma: A Systematic Review and Meta-Analysis

**DOI:** 10.3390/diagnostics12030633

**Published:** 2022-03-04

**Authors:** Angela Santoro, Antonio Travaglino, Frediano Inzani, Patrizia Straccia, Damiano Arciuolo, Michele Valente, Nicoletta D’Alessandris, Giulia Scaglione, Giuseppe Angelico, Alessia Piermattei, Federica Cianfrini, Antonio Raffone, Gian Franco Zannoni

**Affiliations:** 1Unità di Ginecopatologia e Patologia Mammaria, Dipartimento Scienze della Salute della Donna, del Bambino e di Sanità Pubblica, Fondazione Policlinico Universitario A. Gemelli IRCCS, Largo A. Gemelli 8, 00168 Roma, Italy; angela.santoro@policlinicogemelli.it (A.S.); antonio.travaglino.ap@gmail.com (A.T.); frediano.inzani@policlincogemelli.it (F.I.); stracciapatrizia@libero.it (P.S.); damiano.arciuolo@policlinicogemelli.it (D.A.); dr.valente.m@gmail.com (M.V.); ndalessandris@gmail.com (N.D.); scaglionegiulia90@gmail.com (G.S.); giuangel86@hotmail.it (G.A.); alessia.piermattei@policlinicogemelli.it (A.P.); federica.cianfrini@policlinicogemelli.it (F.C.); 2Division of Gynaecology and Human Reproduction Physiopathology, Department of Medical and Surgical Sciences (DIMEC), IRCCS Azienda Ospedaliero-Univeristaria di Bologna, S. Orsola Hospital, University of Bologna, Via Massarenti 13, 40138 Bologna, Italy; anton.raffone@gmail.com; 3Istituto di Anatomia Patologica, Università Cattolica del Sacro Cuore, Largo A. Gemelli 8, 00168 Roma, Italy

**Keywords:** ovarian cancer, chemotherapy, CRS, high grade serous carcinoma, prognosis

## Abstract

Background: chemotherapy response score (CRS) is widely used to assess the response of ovarian high-grade serous carcinoma (HGSC) to chemotherapy and is based on pathological examination of omental specimens. We aimed to assess the prognostic value of CRS assessed on the uterine adnexa. Methods: a systematic review and meta-analysis were performed by searching three electronic databases from 2015 inception to September 2021. We included all studies reporting either hazard ratio (HR) with 95% confidence interval (CI) for progression-free survival (PFS) or primary PFS data, for both adnexal and omental CRS in HGSC. HRs with 95% CI were extracted and pooled by using a significant *p*-value < 0.05. Statistical heterogeneity was assessed by using Higgins’ I2. Results: six studies with 691 HGSC patients were included. Adnexal CRS3 vs. CRS1-2 significantly stratified PFS, with a HR of 0.572 (0.447–0.733; *p* < 0.001). Omental CRS3 vs. CRS1-2 significantly stratified PFS with a similar HR (HR = 0.542; 95% CI 0.444–0.662; *p* < 0.001). Statistical heterogeneity was 0% in both analyses. Conclusions: adnexal CRS significantly stratifies PFS in HGSC and might be used when omental CRS is not assessable.

## 1. Introduction

Ovarian carcinoma represents the most lethal gynaecological malignancy and is ranked as the fifth leading cause of cancer deaths in females [1].

On the basis of morphological features, cellular origin, clinical characteristics, and molecular alterations, ovarian carcinoma can be divided into five main types: (i) high-grade serous carcinoma, (ii) endometrioid carcinoma, (iii) clear cell carcinoma, (iv) mucinous carcinoma, (v) low-grade serous carcinomas. High-grade serous carcinoma (HGSCs) generally harbour TP53 alterations, a pronounced genomic instability, as well as inherited and somatic BRCA1 and BRCA2 mutations [2]. In particular, TCGA research network focused on high grade serous ovarian carcinoma, proposing a molecular classification with prognostic significance without assessing other histotypes [3]. The other ovarian cancer subtypes show mutations in KRAS, BRAF, PTEN, and CTNNB1 (Beta-catenin), as well as a relatively stable karyotype [2,4]. In detail, low-grade serous carcinoma usually shows BRAF and KRAS mutations [2]. β-catenin alterations, microsatellite instability, PTEN, and POLE mutations are frequently observed in endometrioid carcinomas [2,5]. ARID1A mutations, microsatellite instability and *PIK3CA* mutations occur in clear cell carcinomas [2,6]. Mucinous carcinomas are frequently associated with copy-number loss of CDKN2A and KRAS alterations [2]. Recently, considering the crucial role of the TCGA groups in clinical decision making of endometrioid endometrial carcinoma, several studies suggested that the same molecular subgroups could be applied to ovarian endometrioid carcinoma [5].

Despite its infrequent incidence, ovarian cancer can still be considered a prominent public health concern and a clinical challenge in women worldwide [1]. Although the incidence and the mortality rates are showing a slight improvement over the time, falling, respectively, an average 3.3% each year (over 2009–2018) and 2.7% each year (over 2010–2019), ovarian cancer remains one of the significant source of morbidity and mortality in the global population, with an all-stage 5-year relative survival rate equal to 30–50% [6,7].

Among the multiplicity of distinct malignancies that have origin in the ovarian site, HGSC predominates in the clinical and pathological setting, accounting for 70% of all ovarian cancers. HGSC is characterized by rapid growth and early spread to other organs in the peritoneal cavity [1,2]. Although several biomarkers have been assessed to stratify the risk and guide the management of HGSC, it remains a highly lethal malignancy [8,9,10,11,12,13].

The current therapy landscape for tubo-ovarian HGSC is dominated by primary surgical debulking, followed by adjuvant post-operative generally platinum based chemotherapy, being the standard of care for almost 40 years [14]. The choice to proceed with neo-adjuvant chemotherapy (NACT) depends on the impossibility of a surgical radical cytoreduction and/or on medical considerations. However, NACT is increasingly used in western countries due to the reported similar survival outcomes despite lower surgical morbidity [15,16]. Interval debulking surgery (IDS) provides an opportunity for histopathological evaluation of tubo-ovarian HGSC response to NACT and for clinical assessment of prognostic risk. The chemotherapy response score (CRS) represents a simple and reproducible scoring system, developed and validated by Bohm et al., based on post-therapy evaluation of the tumoural architecture and microenvironment in omental site, with significant correlation to progression free survival (PFS), and mixed results for overall survival (OS) [17]. Initially, a six-tier scoring system based on omental and adnexal residual tumour cells was proposed as follows: CRS0-absent tumour response (no fibroinflammatory changes, no evidence of chemotherapy response) with viable tumour only; CRS1- minimal fibroinflammatory changes, mainly viable tumour; CRS2- minor (focal or diffuse) regression-associated fibroinflammatory changes, extensive residual tumour; CRS3- extensive fibroinflammatory changes with focal residual tumour cells; CRS4- extensive fibroinflammatory changes with minimal residual tumour; CRS5- no residual tumour identified. This score, when applied in the omentum, showed a statistically significant correlation with progression free survival (PFS) and overall survival (OS); however, when CRS was evaluated in the adnexa, it did not correlate with the prognosis. Furthermore, authors proposed a simpler three-tier system: (i) CRS1: minimal tumour response; (ii) CRS 2: moderate tumour response, with residual neoplastic foci easily identifiable; (iii) CRS 3: complete or near-complete response, with no residual neoplastic cells or minimal irregularly scattered tumour cells up to 2 mm in maximum size. The three-tier scoring system showed a significant prognostic difference between CRS 1-2 and CRS3 gropus improved the interobserver reproducibility. Therefore, the three-tiered CRS has been included into the International Collaboration on Cancer Reporting (ICCR) and the College of American Pathologists (CAP) guidelines for histopathologic reporting of ovarian carcinoma [18]. From its ideation, other studies have confirmed and reinforced the concept of CRS (both in a three- and two-tiered form) as a prognostic guide in clinical decision-making [19,20,21,22,23,24,25] and a pathological parameter able to identify patients at risk for platinum resistant disease [17,19].

Different results, ranging from no correlation to correlation with only PFS to correlation with both PFS and OS, have been reported when the CRS is applied on the adnexa [17,19,20,26,27,28]. Currently, only the three-tier Böhm’s omental CRS is recommended by the main oncological guidelines [18]. By this systematic review and meta-analysis, we aimed to define the prognostic impact of pathological response to NACT in omental and in the ovarian site.

## 2. Materials and Methods

Study methods were defined a priori based upon previous meta-analyses [29,30]. Each review step (electronic search, study selection, data extraction, risk of bias assessment, data analysis) was performed by two independent authors; all authors consulted at the end of each step to assess the adequacy of the work done until then. The review was reported according to the PRISMA guidelines [31].

### 2.1. Electronic Search and Study Selection

The aim of the present study was to compare for the first time by a systematic re-view and meta-analysis the impact of pathological response to NACT in omental site and in the primary site of tumour (ovarian residual disease), in particular defining the adnexal CRS prognostic value.

Three electronic databases (PubMed, Scopus, and ISI Web of Science) were searched from January 2015 (year of publication of Bohm’s study) to September 2021. Several combinations of the following text words were used: ovarian; ovary; adnexa; CRS; chemotherapy response score; high-grade serous; omentum; omental. Reference lists of relevant studies were also assessed.

All studies that assessed CRS in both omental and adnexal pathological specimens were considered. The *inclusion criteria* were: hazard ratio with 95% confidence interval (CI) for progression-free survival (PFS), extractable for both omental and adnexal CRS; data available for CRS3 vs. CRS1-2 (e.g., HRs for CRS1 vs. CRS 2–3 were not considered). *Exclusion criteria* were (defined a priori) were: sample size <20; reviews; overlapping patient data.

### 2.2. Data Extraction

PICO27 of our study were: P (population) women with HGSOC undergoing neoadjuvant chemotherapy; I (intervention, risk factor) was a CRS3; C (comparator) was a CRS1 or 2; O (outcome) was PFS. HR with 95% CI was extracted from primary studies or calculated by primary data (if available).

### 2.3. Risk of Bias Assessment

According to the QUADAS-228, the risk of bias within studies was assessed in 4 domains: (1) patient selection (selection criteria and period of enrolment reported); (2) index test (adequate reporting of pathological criteria); (3) reference standard (adequate reporting of survival outcomes); (4) flow and timing (median follow-up ≥ 2 years). The risk of bias was categorized as “low”, “high”, or “unclear”, as previously described [31,32].

### 2.4. Data Analysis

HR with 95% CI was calculated from primary data by using Cox regression survival analysis; CRS1-2 was used as reference. If a primary study used CRS3 instead of CRS1-2, as reference, HR was calculated as 1 divided by HR, the lower limit was calculated as HR divided by upper limit, and the upper limit was calculated as HR divided by lower limit. HRs with 95% CI from all studies were pooled by using both a fixed and a random effect model. Statistical heterogeneity was quantified by using Higgins’ I2, as previously described [29,30]. The risk of bias across studies (publication bias) was assessed by using a funnel plot of standard error by logHR. Statistical analyses were performed by using SPSS 19.0 package (SPSS Inc., Chicago, IL, USA) and Comprehensive Meta-Analysis (Biostat, 14 North Dean Street, Englewood, NJ 07631, USA).

## 3. Results

Six studies with 691 HGSOG patients were included [17,19,21,26,28]. The flow diagram of study selection is shown in Appendix A. Characteristics of the included studies are shown in Table 1. No particular risks of bias were found.

The subdivision of patients based on adnexal CRS (CRS3 vs. CRS1-2) significantly stratified PFS with a HR of 0.572 (0.447–0.733; *p* < 0.001) (Figure 1). Statistical heterogeneity among studies was null (I2 = 0%). The funnel plot was symmetric, indicating no significant risk of publication bias (Figure 2).

The subdivision of patients based on omental CRS (CRS3 vs. CRS1-2) significantly stratified PFS, with a similar HR (HR = 0.542; 95% CI 0.444–0.662; *p* < 0.001) (Figure 3). There was neither statistical heterogeneity among studies (I2 = 0%) nor significant risk of publication bias (symmetric funnel plot) (Figure 4).

### Strengths and Limitations

The present review and meta-analysis corroborates the key role of CRS in the assessment of ovarian high-grade serous carcinom, in particular, confirming the prognostic role of adnexal CRS. However, the main limitation of the study is represented by the small number of scientific papers that investigated this topic. In this perspective, further studies are warranted to determine if the CRS (both omental, adnexal, or combined) could be a clinical chance for patient management and personalized therapeutical approaches.

## 4. Discussion

The scientific community started to investigate the relationship between chemotherapy and ovarian cancer, following the paper by McCluggage, in 2002, where author highlighted the histological regressive features in ovarian carcinoma after NACT [33]. In this paper, authors demonstrated significant morphological alteration in both the epithelial and stromal component following chemotherapy. Neoplastic cells, following NACT, were arranged in small groups or in single cells, showing nuclear enlargement and hyperchromasia; cytoplasm was intensely eosinophilic, vacuolated, or with foam-cell changes. Stromal modifications included fibrous changes, inflammation, foamy histiocytes, cholesterol cleft formation, fat necrosis, dystrophic calcifications, and psammoma bodies. Moreover, mitotic figures were often less prominent. Based on these findings, authors concluded that a correct nosological classification of ovarian carcinoma histotypes following chemotherapy can be extremely challenging. However, immunohistochemical analyses can aid in the identification of minimal residual neoplastic cells.

Lately, several studies attempted to assess the prognostic value of morphological alterations observed in ovarian carcinoma following NACT [19,20,21,22,23,24,25,26,27,28].

While the histopathologic response to NACT, in terms of omental CRS, is standing out as a powerful prognostic indicator in patients affected by tubo-ovarian HGSC, the experience with the use of adnexal CRS to stratify EOC patients’ outcomes is still limited by small series and conflicting data.

In the article by Böhm et al. [17], when omental and adnexal CRS were evaluated within a single patient, a better omental score was observed in 41% of patients, equal scores in 43% of patients, and lower omental scores in 16% of patients. However, the omental CRS showed a statistically significant correlation with prognosis, while the adnexal CRS did not show any correlation. Therefore, authors concluded that adnexal residual disease after NACT was difficult and less reproducible to score, without significant correlation with outcome. These findings, differently from all other cancers in which chemotherapy response is evaluated on the primary tumour site, demonstrated, for the first time, that omentum represents the most prognostically relevant disease site for chemotherapy response pathological assessment in ovarian cancer.

Similarly, Singh et al. [20] reported that the prognostic significance of CRS was restricted to the omentum, while, when applied to the adnexal sites, CRS showed no significant association with prognosis (PFS or OS).

However, in a recent study by Santoro et al. [26], where the CRS system was validated in a cohort of 161 patients, the authors demonstrated, for the first time, a statistically significant prognostic stratification of patients based on the adnexal CRS. In detail, regarding adnexal residual disease, significant differences in PFS were observed between CRS1, CRS2, and CRS3 patients, both in univariate and in multivariate analyses. Similarly, Lawson et al. [27] observed that the adnexal three-tiered CRS and modified two-tier score (CRS1-2 vs. CRS3) systems correlate with PFS but not with OS. Authors also examined a combined omental and adnexal scoring system, showing a significant correlation with PFS but no correlation with OS.

These findings are keeping with Michaan et al. [28] and Dizel et al. [19] analyses, which documented significant association between adnexal CRS and/or combined CRS and PFS but not with OS.

In detail, in the study by Michaan et al., CRS3 was more frequently observed in omental tissues compared with adnexal sites, indicating a higher susceptibility to chemotherapy in the omentum [28]. However, CRS3, from both omental and ovarian sites, was significantly related to longer progression-free survival. Moreover, Ditzel et al. [19] showed that the adnexal CRS had statistically significant correlation not only with PFS, but also with OS, although statistical significance was lost after an online-training program on CRS evaluation on adnexal sites. Therefore, authors concluded that online training may not be easily applicable to the adnexa.

The present work represents the first systematic review and meta-analysis, performed on 6 studies with 691 ovarian HGSC patients, confirming the prognostic role of adnexal CRS. We reported that adnexal CRS3 vs. CRS1-2 significantly stratified PFS, with a HR of 0.572 (0.447–0.733; *p* < 0.001), as well as omental CRS3 vs. CRS1-2, significantly stratified PFS with a similar HR (HR = 0.542; 95% CI 0.444–0.662; *p* < 0.001).

These findings suggest that the use of the CRS may not be limited to the omentum, and grading neoplastic response is also possible in adnexal site when omental CRS is not assessable. We must keep in mind that the response of tubo-ovarian HGSC to NACT follows a specific sequential pattern [26,27,28], with regions first involved by tumour (ovaries) being the last to have a complete response, due the higher concentration of disease with multiple neoplastic clones. Of interest, when comparing the CRS scores in the omentum and adnexal site in the Lawson et al. [27] study, the CRS was similar in 47.9% of cases, while the omental score was greater in 39.5% of cases. Similarly, Böhm et al. [17] had 43% of cases with equivalent CRS in the omentum and adnexa, and 41% of cases had a CRS higher in the omentum. These findings confirm the ovaries as the reservoir of drug resistant clones and underline that important prognostic information in patients with HGSC, after NACT, could also be obtained by sampling and assessing the amount of residual tumour in the adnexal site.

Currently, the use of the CRS is limited only to determine disease progression and patient prognosis. Recently, several studies have investigated the possible associations between CRS and radiological and biochemical response, surgical residual disease, laparoscopic score, tumour immune profile, microenvironment, BRCA status, and molecular classification [17,22,27,34,35]. However, the most clinically useful aspects of the omental CRS are represented by its ability to predict cases at risk for early relapse and its association with platinum resistant disease, although opposing data have been reported in literature [17,22,27]. Moreover, although not confirming a correlation between omental CRS and prediction of platinum resistant disease, Lawson et al. [27] stated that the adnexal CRS was significant in predicting platinum-based chemotherapy resistant disease. Moreover, recent studies highlighted the potential role of β-catenin and Aquaporin-1 (AQP1) in serous ovarian carcinoma chemoresistance [36].

In this regard, the Wingless-related integration site(Wnt)/β-catenin pathway has emerged as a key regulator in many steps of ovarian cancer development, including cell proliferation, cancer stem cells survival, metastasis, and chemoresistance [4,36]. Aberrant immunohistochemical expression of β-catenin (nuclear/cytoplasmatic) has also been demonstrated as a surrogate for β-catenin gene (CTNNB1) mutations in ovarian carcinoma [4,36,37].

AQP1 is a small trans-membrane water channel protein involved in cell proliferation, adhesion, and motility, as well as in the modulation of serous fluid volumes [38]. Immunohistochemical expression of AQP1 has been demonstrated as a prognostic biomarker in several solid tumours, including mesothelioma, breast cancer, colorectal cancer, brain tumours, prostate adenocarcinoma, lung adenocarcinoma, and ovarian cancer [38,39,40,41].

A recent study, analysed the relationship between AQP1 expression and omental chemotherapy response in serous ovarian carcinoma [40].

In detail, authors demonstrated a statistically significant association between AQP1 expression and poor chemotherapy omental response (CRS1-2), suggesting that AQP1 could represent a predictive biomarker of platinum resistance in ovarian cancer.

In this perspective, further studies are warranted to determine if the CRS (both omental, adnexal, or combined) could be a clinical opportunity for patient management and personalized treatments, being able to discriminate both optimal candidates for additional adjuvant therapies, such as immunotherapeutic agents, poly (adenosine diphosphate–ribose) polymerase (PARP) inhibitors, and more-resistant patients, which should be quickly assigned to clinical trials of new therapeutical regimen.

## 5. Conclusions

Currently, the three-tiered omental CRS system allows for a clinically meaningful evaluation of the pathological response, in women with HGSC, after undergoing NACT. Overall, this systematic review and meta-analysis confirms that the omental CRS correlates with PFS when used as a modified two-tier system. We also showed that the CRS, when measured on the adnexa, correlates with PFS, when graded on the modified two-tier system. These findings support the idea that the adnexal CRS is also a reproducible and clinically relevant system, since it could be helpful in defining systemic treatment after neoadjuvant therapy and interval cytoreduction. Therefore, we retain that cancer reporting protocols (such as those by the College of American Pathologists and the International Collaboration on Cancer Reporting) should continue to consider the use of the three and/or two-tier omental CRS, as well as possibly expand the CRS evaluation to the adnexa in the pathological report. However, all the news in guidelines should be based on comprehensive gene profiling and well-designed randomized clinical trials

## Figures and Tables

**Figure 1 diagnostics-12-00633-f001:**
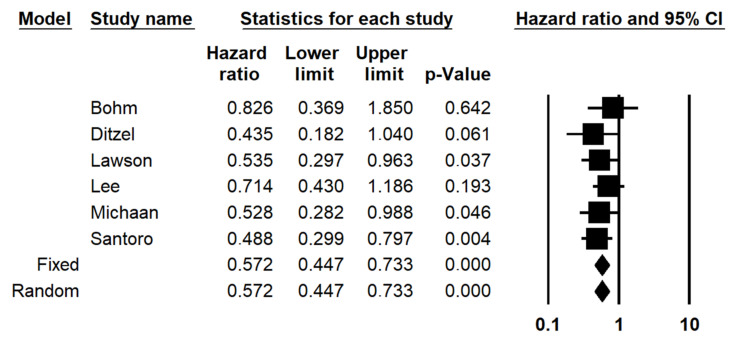
Forest plot of the hazard ratio (HR) for progression-free survival in ovarian high-grade serous carcinoma (adnexal CRS1 vs. CRS2-3).

**Figure 2 diagnostics-12-00633-f002:**
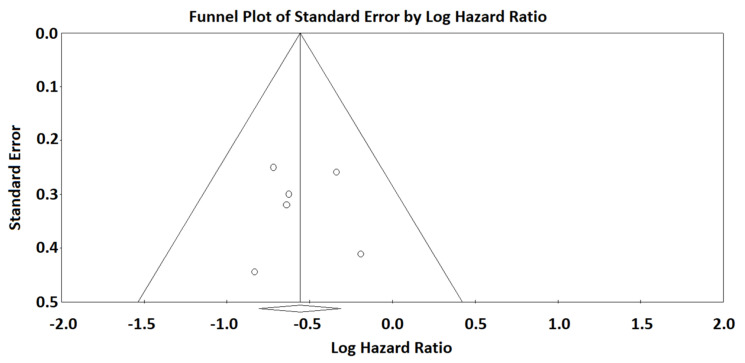
Funnel plot of standard error by logHR for the analysis of adnexal CRS. The vertical line with the diamond sign at the bottom indicates the logarithm of the HR for progression-free survival. The symmetry of the funnel plot indicates that there is no significant risk of publication bias.

**Figure 3 diagnostics-12-00633-f003:**
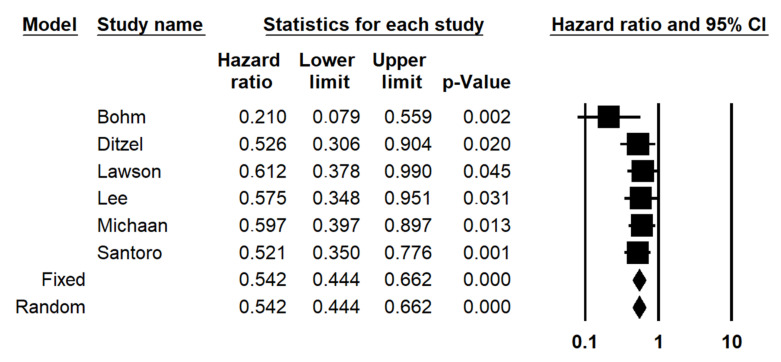
Forest plot of the hazard ratio (HR) for progression-free survival in ovarian high-grade serous carcinoma (omental CRS1 vs. CRS2-3).

**Figure 4 diagnostics-12-00633-f004:**
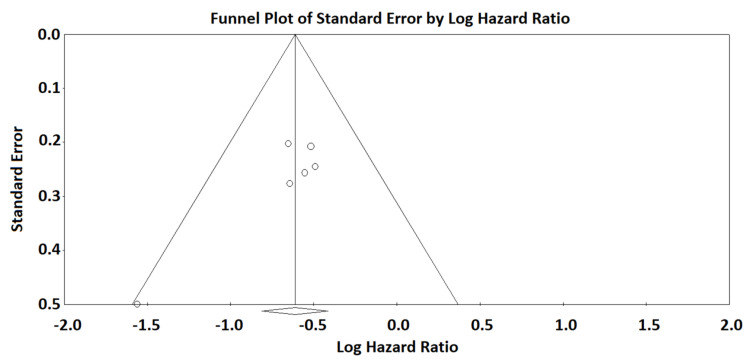
Funnel plot of standard error by logHR for the analysis of omental CRS. The vertical line with the diamond sign at the bottom indicates the logarithm of the HR for progression-free survival. The symmetry of the funnel plot indicates that there is no significant risk of publication bias.

**Table 1 diagnostics-12-00633-t001:** Characteristics of the included studies.

Study	Country	Sample Size	Period of Enrollment
Bohm 2015	UK (test cohort)	62 (test cohort)	2009–2014
Lee 2017	Korea	110	2006–2014
Ditzel 2018	Massachusetts (USA)	68 (59 adnexal)	2005–2012
Michaan 2018	Korea	132	2009–2014
Santoro 2019	Italy	161	2014–2017
Lawson 2020	Texas (USA)	158	2013–2018

## Data Availability

Not applicable.

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
