# Peer review of "Prognostic Value of Chemotherapy Response Score (CRS) Assessed on the Adnexa in Ovarian High-Grade Serous Carcinoma: A Systematic Review and Meta-Analysis"

_diagnostics, 2022, doi:10.3390/diagnostics12030633_

Round 1

Reviewer 1 Report

The chemotherapy response score (CRS) is increasingly emerging as an important histopathological tool to evaluate chemotherapy response in women with ovarian high-grade serous carcinoma, one of the most common tubal/ovarian malignant tumors that often unveils at a late stage. Santoro et al., in the present work performed a systematic review and meta-analysis to further corroborate the key role of CRS in the assessment of ovarian high-grade serous carcinoma. The analysis is interesting, even though the authors should also include a little paragraph on the strengths and limitations of their study.

Overall, the manuscript is well written and organized.

Minor comments:

-pag. 1, line 41: open abbreviation HGSCs and use it at pag. 2, line 63.

-pag. 3, line 111: here the authors should provide some references to support their statement.

-pag 3: in the Materials and Methods section, I would suggest to combine sub-section 2.1 and sub-section 2.2. Under this section, I would also recommend the authors to provide a better description of the search strategy, and the eligibility criteria applied (e.g., inclusion and exclusion criteria).

-Fig.1 and Fig. 3: The authors should specify to what the area of the squares is proportional to, and correspondingly provide more information in the Figure legend.

-Fig. 2 and Fig.4: Increase the font and provide more information in the Figure legend.

Author Response

Manuscript ID: diagnostics-1616781

Type of manuscript: Systematic Review

Title: Prognostic Value of Chemotherapy Response Score (CRS) Assessed on the Adnexa in Ovarian High-Grade Serous Carcinoma: A Systematic Review and Meta-Analysis

Corresponding Author: Gian Franco Zannoni

Co-Authors: Angela Santoro, Antonio Travaglino, Frediano Inzani, Patrizia Straccia, Damiano Arciuolo, Michele Valente, Nicoletta D'Alessandris, Giulia Scaglione, Giuseppe Angelico, Alessia Piermattei, Federica Cianfrini, Antonio Raffone

Submitted to section: Pathology and Molecular Diagnostics

Dear Editor,

Thank you for giving us the chance to enhance our manuscript

Below is each question raised by the Reviewer, followed by our response.

We submitted the revised manuscript, underlining the corrections.

Reviewer 1

Q: The chemotherapy response score (CRS) is increasingly emerging as an important histopathological tool to evaluate chemotherapy response in women with ovarian high-grade serous carcinoma, one of the most common tubal/ovarian malignant tumors that often unveils at a late stage. Santoro et al., in the present work performed a systematic review and meta-analysis to further corroborate the key role of CRS in the assessment of ovarian high-grade serous carcinoma. The analysis is interesting, even though the authors should also include a little paragraph on the strengths and limitations of their study.

Overall, the manuscript is well written and organized and will be of high interest for the scientific community.

A: Thank you for the positive comments. We have added in results section the advised paragraph

Minor comments:

Q: pag. 1, line 41: open abbreviation HGSCs and use it at pag.2, line 63.

A: We have done it

Q: pag. 3, line 111: here the authors should provide some references to support their statement.

A: We have done it

Q: pag 3: in the Materials and Methods section, I would suggest to combine sub-section 2.1 and sub-section 2.2. Under this section, I would also recommend the authors to provide a better description of the search strategy, and the eligibility criteria applied (e.g., inclusion and exclusion criteria).

A: We have done it

Q: Fig.1 and Fig. 3: The authors should specify to what the area of the squares is proportional to, and correspondingly provide more information in the Figure legend.

A: We thank the Reviewer for the comment. Please note that all squares have the same area.

Q: Fig. 2 and Fig.4: Increase the font and provide more information in the Figure legend.

A: We thank the Reviewer for the advice. We now increased the font and provided more information in the Figure legend.

We look forward to hearing from you.

Sincerely,

Gian Franco Zannoni, M.D. (for all authors)

Reviewer 2 Report

These results could be helpful in defining systemic treatment after neoadjuvant therapy and interval cytoreduction. The news in guidelines should be based on comprehensive gene profiling and well-designed randomized clinical trials

Author Response

These results could be helpful in defining systemic treatment after neoadjuvant therapy and interval cytoreduction. The news in guidelines should be based on comprehensive gene profiling and well-designed randomized clinical trials

A: We agree to the Reviewer’s comment; we have also added it in the Conclusions

Reviewer 3 Report

Very interesting and well written manuscript. The systematic review is well conducted and the presented information is useful.  

The fact that adnexal CRS significantly stratifies PFS in HGSC and might be used when omental CRS is not assessable is very valuable. In detail, authors demonstrated a statistically significant association between AQP1 285 expression and poor chemotherapy omental response (CRS1-2), suggesting that AQP1 286 could represent a predictive biomarker platinum resistance in ovarian cancer. The systematic review is well conducted and the presented information is useful.  

Author Response

Very interesting and well-written manuscript. The systematic review is well conducted and the presented information is useful. 

The fact that adnexal CRS significantly stratifies PFS in HGSC and might be used when omental CRS is not assessable is very valuable. In detail, authors demonstrated a statistically significant association between AQP1 285 expression and poor chemotherapy omental response (CRS1-2), suggesting that AQP1 286 could represent a predictive biomarker platinum resistance in ovarian cancer. The systematic review is well conducted and the presented information is useful. 

A: Thank you for the positive comments